# An Energy-Efficient Coverage Enhancement Strategy for Wireless Sensor Networks Based on a Dynamic Partition Algorithm for Cellular Grids and an Improved Vampire Bat Optimizer

**DOI:** 10.3390/s20030619

**Published:** 2020-01-22

**Authors:** Xiaoqiang Zhao, Yanpeng Cui, Zheng Guo, Zhanjun Hao

**Affiliations:** 1School of Communication and Information Engineering, Xi’an University of Posts and Telecommunications, Xi’an 710121, China; zxq7703@126.com (X.Z.); gz95428@126.com (Z.G.); 2Shaanxi Key Laboratory of Information Communication Network and Security, Xi’an University of Posts and Telecommunications, Xi’an 710121, China; 3School of Computer Science and Engineering, Northwest Normal University, Lanzhou 730070, China; zhanjunhao@126.com

**Keywords:** wireless sensor networks, coverage effect, dynamic partition, cellular grid, energy consumption, task distributing, improved vampire bat optimizer

## Abstract

Sensor nodes perform missions based on the effectual invariable coverage of events, and it is commonly guaranteed by the determinate deployment for sensor nodes who deviate from the optimum site frequently. To reach the optimal coverage effect with the lowest costs is a primary goal of wireless sensor networks. In this paper, by splicing the sensing area optimally with cellular grids, the best deployment location for sensors and the required minimum number of them are revealed. The optimization problem of coverage rate and energy consumption is converted into a task assignment problem, and a dynamic partition algorithm for cellular grids is also proposed to improve the coverage effect when the number of sensors is variable. Furthermore, on the basis of solving the multi-objective problem of reducing and balancing the energy cost of sensors, the vampire bat optimizer is improved by introducing virtual bats and virtual preys, and finally solves the asymmetric assignment problem once the number of cellular grids is not equal to that of sensors. Simulation results indicate that the residual energy of sensors during redeployment is balanced notably by our strategy when compared to three other popular coverage-enhancement algorithms. Additionally, the total energy cost of sensor nodes and coverage rate can be optimized, and it also has a superior robustness when the number of nodes changes.

## 1. Introduction

With the rapid development of wireless communication technology, embedded computing technology, sensor technology, and microelectronic technology, wireless sensor networks (WSNs) which bring low-power, low-cost, distributed and self-organizing features to information perception have emerged at this historic moment. They have greatly changed the way humans interact with nature and established a bridge between the information and the physical world [1]. As a kind of self-organized network formed by low-power microsensor nodes with the ability of sensing, data processing and storing in wireless communication, WSNs, which are called one of the most influential technologies in the 21st century, have become one of the most popular research fields due to their wide application in military applications, environmental monitoring and natural disaster prediction, smart home, medical and health care, and even outer space exploration [2].

Coverage control and node deployment, which determine the ability to monitor the surrounding physical world and the quality of service (QoS) of WSNs, are one of the core issues in WSNs research. How to improve the QoS of sensing, monitoring, communication and other services by coverage control and node deployment strategy, and eventually prolong the life-cycle of network, has been an important challenge of WSNs in recent years. Sensor nodes perform missions based on the effectual invariable coverage of events [3], hence the network reliability and monitoring quality of WSNs cannot be improved unless monitoring events can effectively be detected by sensor nodes [4]. Given the energy limitations of sensor nodes and the harsh, complex sensing area, the initial layout is usually formed by random dispersal, such as an airdrop. However, some problems will be caused due to the irregular deployment, for example, the low-density deployment will lead to monitoring blind areas, and the high-density deployment will make the monitoring areas of sensor nodes overlap with each other, which will lead to the mutual competition on the shared channel and communication interference between nodes, that not only affects the reliability of data transmission, but also causes large energy costs. Accordingly, the mobile sensors are redeployed to enhance their coverage effect.

Resource constraints are another major feature of WSNs, and also a major bottleneck that hinders large-scale redeployment. Given that sensors are small embedded system devices, which are usually equipped with radio transceivers, microcontrollers, and energy supplies (usually batteries), their energy supply is often difficult to achieve due to the limitations of WSN application scenarios. This can cause irreparable disasters in an entire network once the nodes fail due to energy exhaustion. Therefore, it is necessary to reduce the node energy consumption as much as possible and prolong the working cycle to the greatest extent [5]. The movement of sensor nodes and signal transmission between them are two main aspects of energy consumption during redeployment, and the proportion of the former is higher than that of the latter for mobile sensors. Accordingly, in addition to the coverage enhancement, the optimization of moving distance is also an essential factor for mobile WSNs [6].

## 2. Related Works

Sensor nodes are abstracted as particles in force field by a virtual force algorithm (VFA), which is the most popular and primitive algorithm when solving problems of coverage control. The predecessor of VFA is virtual potential field, which was first used to solve the problem of path planning and obstacle avoidance for robot according to the virtual force defined by the environment around. VFA was used to solve the coverage enhancement problem of WSNs by Zou in [7], where all sensors are steered by virtual attraction and repulsion and carry out virtual instead of physical moving task for each iteration. When the sensors are closely deployed to each other, the virtual repulsion between them can reduce redundant coverage, while the virtual gravity will make the global coverage more uniform once they are too far apart. In addition, obstacles and priority monitoring areas also act as repulsive and gravitational sources respectively. VFA is implemented in cluster heads with higher energy, and all sensor nodes finally move linearly to the destination to enhance the coverage effect after the coverage rate or the number of iterations exceeds the threshold. VFA has the advantages of simple implementation, flexible application and obvious optimization effect, hence it has been improved by a large number of researches [8,9,10,11]. Particle swarm optimization (PSO) and VFA are combined into a new algorithm called virtual force-directed particle swarm optimization (VFPSO) in [9], by introducing the virtual force to the update of moving speed of particles, the individual’s historical optimal position, the global optimal solution and the virtual force drive the movement of particle together. The authors of [11] embeds the definition of VFA and the Lévy flight strategy into the updating formula of grey wolf’s position, which can accelerate the convergence speed, and it has a better performance in coverage and moving distance. However, the algorithm based on VFA only considers the optimization of coverage effect and ignores the optimization of moving distance. In addition, VFA cannot achieve the best coverage effect in some cases even if the number of sensors is sufficient.

As one of the research fields of artificial intelligence, metaheuristic algorithms are an optimization tool for solving large-scale optimization problems based on the idea of sacrificing solution accuracy for solution efficiency, which is based on the division of labor and self-organization behavior of biological groups [12,13,14]. A novel modified GWO with Lévy flight (LF) is proposed in [15], where the whole wolf group is divided into four groups (α, β, δ and ω), and the first three groups are the best adaptive wolves guiding other wolves (ω) to search for the target, which is similar to GWO. Compared with the original GWO and other heuristic algorithms, the optimization performance of LGWO was effectively improved in tackling engineering optimization tasks, which is mainly attributed to the integration of LF and greedy selection strategy with improved hunting methods. In addition, by establishing the relationship between wolves and cluster heads and dynamically setting the weights of different grades of wolves, the authors in [16] improved GWO and finally applied it to optimize the clustering routing protocol of WSNs. A Harmony Search (HS) algorithm was proposed in [17] to improve the connectivity of the non-uniform density WSNs to increase network connectivity and enhance coverage effect. Based on both probabilistic and binary monitoring models, by controlling the sensing range and position of mobile sensor nodes and dynamically adjusting the number of activated sensors, a Centralized Immune-Voronoi deployment Algorithm (CIVA) was proposed in [18] to enhance coverage effect. In order to solve the coverage enhancement problem and optimize the maximum service life of sensor nodes in WSNs, a distributed algorithm based on a generalization of the Cellular Automata concept called Graph Cellular Automata (GCA) was proposed in [19]. Particle swarm optimization (PSO) [20] is widely used in the process of multi-modal optimization because of its simple parameters, strong optimization ability, fast convergence speed and low time complexity [21]. The particle swarm optimization (PSO) has been used and improved by a number of surveys [22,23,24] to enhance cover effect in WSNs. The authors in [25] combine PSO with Voronoi to optimize the coverage enhancement of WSN, PSO is used to find the best deployment location, and Voronoi is responsible for evaluating the excellence of the scheme. In [26], the artificial bee colony algorithm and PSO are combined to optimize the efficient coverage strategy of sensor nodes, including the rotation working, and the multi-objective function about the final coverage rate and the number of sensors. How to use the collaboration of data fusion and deployed sensors to enhance the performance of coverage is ignored in most existing surveys and analysis studies, the authors of [27] present a comprehensive summary and classification of coverage optimization strategies based on data fusion. Based on the learning automata and cellular learning automata models, an overall research framework is proposed, including the confident information coverage (CIC)-based sensor scheduling for CIC-based deployment for minimizing network deployment cost, maximizing network lifetime, and CIC-based coverage hole detection and healing. In order to avoid the negative impact of coverage holes on network QoS, by using the CIC model and taking full account of the energy consumption and communication capabilities of the sensor nodes, [28] proposes an EICICHD for the coverage hole detection problem from the perspective of energy saving. Based on the correlation of the internal spatial distribution of the monitored variables and the cooperative perception between adjacent sensors, the efficiency of CIC hole detection is improved, and the positions and number of CIC holes are effectively located and determined.

Optimization strategies based on task assignment have the ability of quick convergence in addition to escaping from local optima. A group-based mobile sensor dispatch algorithm was proposed in [29], which groups the monitoring events by their positions. The sensors are dispatched to each cluster of events to reduce and balance energy consumption during movement, and it greatly extends the lifetime. Aimed at guaranteeing the task completion instantaneously and extending the life-cycle of network, the authors in [30] presented an energy-efficiency node scheduling algorithm based on game theory for WSNs, and the payoff function includes both the residual energy and local task load of the sensor nodes. A Hungarian algorithm (HA) [31] is used to solve the NP-hard problem of deterministic coverage enhancement with small time complexity [32], that is, the shortest moving scheme between the initial position of each sensor node and the position to be deployed is determined by the maximum matching algorithm of bipartite graph [33,34]. The above algorithms based on task assignment all ignore the optimization of reducing the maximum energy cost of sensors and balancing the residual energy, which are exactly the keys that affect the life-cycle of WSNs.

The related works reveals that the derivative algorithms of VFA and GWO have relatively superior performance in solving the problem of sensor coverage enhancement, but they have limitations such as the upper limit of coverage optimization ability and the difficulty in optimizing mobile energy consumption, hence a new strategy is proposed in this paper and compared with VFA, VFPSO and LGWO which are the most representative algorithms. The central contributions of this paper are as follows:(1)We present a stacking strategy based on cellular grid (SSBCG) for splicing the two-dimensional sensing area optimally with the length and width given.(2)A cellular grids dynamic partitioning algorithm (CGDPA) is proposed to dynamically adjust the size of the cellular grid based on the actual number of sensor nodes to optimize the coverage effect when the number of sensors changes.(3)The optimization problem of coverage enhancement and energy consumption is converted into a task distributing problem of assigning cellular grids for sensor nodes. We improved the vampire bat optimizer (IVBO), which has been introduced and discussed in [35], to solve the asymmetric competition problem by introducing virtual bats and virtual preys, namely not only the multi-objective problem of minimizing and balancing the energy cost of nodes, but also the asymmetric assignment problem once the number of cellular grids is not equal to that of sensor nodes.(4)Simulation experiments are performed with MATLAB, and the proposed strategy is compared with VFA, VFPSO and LGWO, and the reasons for their performance differences in energy cost of nodes and final coverage rate are revealed and discussed.

The structure of the paper is as follows. The related concepts of two-dimensional deterministic coverage problem are described in Section 3, mainly including energy consumption model, coverage model and mathematical optimization model of deterministic coverage. In Section 4, a stacking strategy based on cellular grid, an improved vampire bat optimizer for asymmetric assignment problem, and a cellular grids dynamic partitioning algorithm are presented to solve the problem of deterministic coverage enhancement when sensor nodes change. Simulation analysis and discussions of the reasons that causing the performance differences of VFPSO, VFA, LGWO and the proposed strategy are given and in Section 5 and Section 6, respectively. Ultimately, we summarize the main contributions of this paper in Section 7.

## 3. Problem Statement

### 3.1. Two-Dimensional Coverage Model of Sensors

The sensing area Ω can be symbolized by a two-dimensional area comprising K discrete points. Note the sensing range of the sensor node in Ω as Θ, whose radius denotes the perceived range of the sensor node. $G_{j}$ can be monitored by $S_{i}$ once the condition $d_{i,j}\leq R_{S}$ is met, where $S_{i}$ represents the i-th sensor node and its location $\left(x_{S_{i}},y_{S_{i}}\right)$, $G_{j}$ represents the centroid of the j-th discrete point and its coordinates $\left(x_{G_{j}},y_{G_{j}}\right)$, the sensing radius of all sensors and the distance between $S_{i}$ and $G_{j}$ are denoted as $R_{S}$ and $d_{i,j}$, respectively. The probability that $G_{j}$ covered by $S_{i}$ can be calculated by:(1)$$p_{i,j}\;=\;\{\begin{array}{c} \;\;1,\;if\;d_{i,j}\leq R_{S}\\ 0,\;otherwise \end{array}.$$

Given that $G_{j}$ may be covered by many sensors simultaneously, the condition that $G_{j}$ has been successfully covered is that $G_{j}$ has been monitored by at least one sensor, and the coverage probability for $G_{j}$ can be calculated by $p_{j}\;=\;1-\prod_{S_{i}\in S_{\alpha }}\left(1-p_{i,j}\right)$, where $S_{\alpha }$ is the sensor set that covers $G_{j}$. Accordingly, the coverage rate (CR) of Ω can be calculated by $\sum_{j\;=\;1}^{K}p_{j}/K$.

### 3.2. Coverage Enhancement

Sensor nodes perform missions based on the effectual invariable coverage of events, which is commonly guaranteed by deterministically redeploying sensor nodes who deviate frequently from the optimum site. To reach the optimal coverage effect with lowest costs is one of the primary goals of WSNs. Assuming that all sensors have the same perceived radius, and can acquire location information and reach any position in Ω, a coverage enhancement strategy can be regarded as the most efficient once it can save the amount of sensors with the cover effect of Ω fully optimized, which is equivalent to seeking a polygon who has the highest efficiency in stacking Ω [36]. The optimal coverage pattern is the regular hexagon with the sensing range $R_{S}$ as its side length [37,38].

### 3.3. Energy Consumption During Coverage Enhancement

Given that the failure time of the node dead firstly is often considered as an important indicator to measure the life-cycle of WSNs, and the movement of sensor nodes and signal transmission between them are two main aspects of energy consumption during redeployment, which means the proportion of the former is higher than that of the latter for mobile sensors, namely the moving distance is a necessary factor for a mobile WSNs. Consequently, in addition to the coverage enhancement, the optimization of energy cost is also an essential factor for mobile WSNs, which is equivalent to the optimization of mobile distance of sensors.

After moving to $G_{j}$, the residual energy of $S_{i}$ is defined as $E_{i,j}\;=\;Eo_{i}-e\times d_{i,j}$, the total energy cost (TEC) of all sensors can be calculated by $e\sum_{i\;=\;1}^{N_{S}}d_{i,j}$, the maximum energy cost (MEC) of nodes can be calculated by $\underset{i\;=\;\left\{1,2,\cdots ,N_{S}\right\}}{\max}e\times d_{i,j}$, and the uniformity of residual energy (URE) of all sensors can be calculated by $U\;=\;\sqrt{\sum_{i\;=\;1}^{N_{S}}{\left(E_{i,j}-\frac{1}{N_{S}}\sum_{i\;=\;1}^{N_{S}}E_{i,j}\right)}^{2}}$, where $S_{i}$’s initial energy is denoted as $Eo_{i}$, and its energy cost after deploying a movement of 1 m is denoted as e, and $N_{S}$ is the number of sensors.

The redeployment problem can be transformed to a task distributing problem of assigning $N_{C}$ mobile destinations for $N_{S}$ sensor nodes, Figure 1 shows the bipartite graph model of it, whose weight of the edge $<S_{i},G_{j}>$ is equivalent to $d_{i,j}$, $D_{N_{S}\;\times \;N_{C}}$ can be presented as:(2)$$D_{N_{S}\times N_{C}}\;=\;\left[\begin{array}{ccc} d_{1,1} & \cdots & d_{1,N_{C}}\\ \vdots & \ddots & \vdots \\ d_{N_{S},1} & \cdots & d_{N_{S},N_{C}} \end{array}\right].$$

The objective function can be defined as $\min\left(w_{1}f_{1}+w_{2}f_{2}\right)$:(3)$$\{\;\begin{array}{c} f_{1}\;=\;\frac{1}{N}\sum_{i\;=\;1}^{N_{S}}\sum_{j\;=\;1}^{N_{C}}d_{i,j}\times x_{i,j}\;\;\;\;\;\;\;\;\;\;\;\;\;\;\;\;\;\;\\ f_{2}\;=\;\sqrt{\frac{1}{N}\sum_{i\;=\;1}^{N_{S}}\sum_{j\;=\;1}^{N_{C}}{\left(d_{i,j}\times x_{i,j}-f_{1}\right)}^{2}} \end{array}.$$
and the constraint condition is:(4)$$s.t.\{\begin{array}{c} \sum_{i\;=\;1}^{N_{S}}x_{i,j}=1,\;j=1,2,\ldots ,N_{C}\\ \sum_{j\;=\;1}^{N_{C}}x_{i,j}=1,\;i=1,2,\ldots ,N_{S}\\ x_{i,j}=\{\begin{array}{c} 1,\;G_{j}\;is\;the\;destination\;of\;S_{i}\\ 0,\;otherwise \end{array} \end{array},$$
where $f_{1}$ and $f_{2}$ are the cost functions about TEC and URE of sensors during movement; $w_{1}$ and $w_{2}$ are the weights of $f_{1}$ and $f_{2}$, respectively.

## 4. Proposed Algorithm

### 4.1. Stacking Strategy Based on Cellular Grid

With the intention of maximizing the sensing range of all sensors, it is necessary to specify the deployment location of each sensor, hence a stacking strategy based on cellular grids (SSBCG) is proposed.

As the arrow shows in Figure 2, the perceived radius of sensor nodes is represented as the radius of the circumcircle of the cellular grid, which is recorded as $R_{C}$. Given that there is a geometric relationship of $\left|\vec{BL_{2}}\right|\;=\;\sqrt{3}\left|\vec{DL_{2}}\right|=\;3R_{C}/2$, $\left|\vec{BL_{3}}\right|\;=\;\left|\vec{L_{3}L_{4}}\right|/2=R_{C}/2$, and $\left|\vec{BL_{4}}\right|\;=\;\sqrt{3}\left|\vec{L_{3}L_{4}}\right|/2\;=\;\sqrt{3}R_{C}/2$, namely the relationship between $R_{C}$ and stacking interval Δx and Δy, which are as twice the length of $\left|\vec{BL_{2}}\right|$ and $\left|\vec{BL_{4}}\right|$, satisfies $\Delta x=3R_{C}$ and $\Delta y=\sqrt{3}R_{C}$, respectively; the coordinates of the cellular grid B and D, which is the reference grid of the first and second type of cellular grids, are $\left(R_{C}/2,0\right)$ and $\left(2R_{C},\sqrt{3}R_{C}/2\right)$, respectively.

When using cellular grids with a radius of $R_{C}$ to seamlessly stack Ω with L and W as length and width, all cellular grids can be classified into two categories according to their coordinates. The first type of grid is based on the reference grid $\left(R_{C}/2,0\right)$ and is extended in the horizontal and vertical directions with Δx and Δy as the stacking interval, respectively. Accordingly, the centroids and the locations of the first type of cellular grids are denoted as $\left(R_{C}/2+n_{1}\Delta x,n_{2}\Delta y\right),\;n_{i}\in \left\{1,\;2,\;\ldots ,\;N_{i}\right\}$:(5)$$N_{i}=\{\begin{array}{c} ⎣(L-R_{C}/2)/\Delta x+1⎦,\;\;i=1\\ ⎣W/\Delta y+1⎦\;\;\;\;\;\;\;\;\;\;\;\;\;\;\;\;,\;\;i=2 \end{array},$$
where $N_{1}$ and $N_{2}$ are the minimum number of the first type of cellular grids required along axis X and Y, respectively, when using the first type of cellular grids to stack Ω seamlessly. The distance from the centroid of cellular grid B to the edge of Ω along axis X and Y can be expressed as $L-R_{C}/2$ and W due to the coordinates of point B, hence the number of remaining cellular grids required along axis X and Y can be expressed as $\left(L-R_{C}/2\right)/\Delta x$ and W/Δy. The number of all cellular grids required along axis X and Y is $\left[\left(L-R_{C}/2\right)/\Delta x+1\right]$ and ⎣W/Δy+1⎦, respectively, since cellular grid B should be also included. The number of sensor nodes can be economized by rounding down the result.

Analogously, with the same stacking interval as the first type of cellular grids, and regarding the centroid $\left(2R_{C},\sqrt{3}R_{C}/2\right)$ as the reference grid, the centroids and the locations of the second type of cellular grids are denoted as $\left(2R_{C}+n_{3}\Delta x,\sqrt{3}R_{C}/2+n_{4}\Delta y\right),n_{i}\in \left\{1,\;2,\;\ldots ,\;N_{i}\right\}$:(6)$$N_{i}\;=\;\{\;\;\begin{array}{c} ⎣(L-2R_{C})/\Delta x+1⎦\;\;\;\;\;\;\;\;\;\;,\;\;i\;=\;3\\ ⎣(W-\sqrt{3}R_{C}/2)/\Delta y+1⎦\;\;,\;\;i\;=\;4 \end{array}.$$

$N_{3}$ and $N_{4}$ are the minimum number of the second type of cellular grids required along axis X and Y, respectively, when using the second type of cellular grids to stack Ω seamlessly. Therefore, the calculation formula of the minimum number of required cellular grids to stack Ω seamlessly is:(7)$$N_{min}\;=\;N_{1}N_{2}+N_{3}N_{4}.$$

By using cellular grids with a radius of $R_{C}\;=\;5.25\;m$ to stack Ω with a size of 50 m×50 m seamlessly, the minimum number of cellular grids can be calculated by Equations (5)–(7), which is 42. The stacking effect is shown in Figure 3, the blue cellular grids are the first type of grids based on the 1-st grid, and the remaining cellular grids are the second type of grids based on the 7-th grid.

### 4.2. Improved Vampire Bat Optimizer

Based on SSBCG proposed in Section 4.1, the coverage effect can be enhanced by deploying N sensor nodes to the centroids of the N cellular grids. However, how the moving destination of each sensor node should be allocated has not been resolved, which is related to the movement trajectory of the nodes and even the energy cost during redeployment. Inspired by the vampire bat’s egoism and altruism, we have introduced and discussed vampire bat optimizer (VBO) in [35], aiming at maximizing the benefits of the whole generation of vampire bats and balancing the health status of individual. Eventually, VBO is used for reducing TEC, MEC, and URE of sensors during deployment.

However, VBO can only solve the problem of symmetric assignment, namely the problem of coverage enhancement that the number of sensor nodes is equal to that of mobile destinations. VBO is not applicable once they are not equal. For the case that the number of cellular grids is not equal to that of sensor nodes, we propose an improved VBO (IVBO) to solve the asymmetric competition problem by introducing virtual bats and virtual preys for the asymmetry of competition process. IVBO solves not only the multi-objective problem of minimizing the TEC and URE of sensor nodes, but also the asymmetric assignment problem once the assignment matrix is not square, which has the following steps.

#### 4.2.1. Seeking the Favorite Prey

As a unique blood-eating mammal, the feeding habits of vampire bats vary widely. Whether they are interested in a prey depends not only on their own taste and hunger degree, but also on the blood volume and hunting risk of the target prey. Assuming that the number of bats and prey is $N_{b}$ and $N_{p}$ respectively, and they are not necessarily equal. For convenience, the j-th prey and the i-th bat are denoted as $p_{j}$ and $b_{i}$, respectively. The risk rate of $p_{j}$, the gene of $b_{i}$, and the interest rate of $b_{i}$ in $p_{j}$ are denoted as $r_{j}^{t}$, $g_{i}^{t}$ and $I_{i,j}^{t}$, respectively.

Before the whole generation of vampire bats starts hunting, the $N_{b}$ bats will calculate the income of the intended hunting according to their interest in prey and the risk of capturing the $N_{p}$ prey. For example, $B_{i,j}^{t}\;=\;I_{i,j}^{t}-r_{j}^{t}$ can characterize the benefits of $b_{i}$ to capture $p_{j}$ during the t-th round of hunt, and the benefit matrix is defined as:(8)$$B_{N_{b},N_{p}}^{t}\;=\;\left[\begin{array}{ccc} B_{1,1}^{t} & \cdots & B_{1,N_{p}}^{t}\\ \vdots & \ddots & \vdots \\ B_{N_{b},1}^{t} & \cdots & B_{N_{b},N_{p}}^{t} \end{array}\right].$$

If $N_{b}$ is greater than $N_{p}$, the competition problem is transformed into a symmetrical assignment problem by adding $N_{b}-N_{p}$ virtual preys, which is equivalent to adding $N_{b}-N_{p}$ columns of zero element to the right of matrix $B_{N_{b},N_{p}}^{t}$ and expanding it to a square matrix of $N_{b}\times N_{b}$ as shown in Equation (9). Namely $N_{p}$ bats can capture prey and suck blood eventually with the remaining $N_{b}-N_{p}$ bats starved until the second part of IVBO:(9)$$B_{N_{b},N_{b}}^{t}\;=\;\left[\;\begin{array}{ccc} B_{1,1}^{t} & \cdots & B_{1,N_{p}}^{t}\\ \vdots & \ddots & \vdots \\ B_{N_{b},1}^{t} & \cdots & B_{N_{b},N_{p}}^{t} \end{array}|\;\begin{array}{ccc} 0 & \cdots & 0\\ \vdots & \ddots & \vdots \\ 0 & \cdots & 0 \end{array}\right].$$

If $N_{p}$ is greater than $N_{b}$, the competition problem is transformed into a symmetrical assignment problem by adding $N_{p}-N_{b}$ virtual bats, which is equivalent to adding $N_{p}-N_{b}$ rows of zero element to the top of matrix $B_{N_{b},N_{p}}^{t}$ and expanding it to a square matrix of $N_{p}\times N_{p}$ as shown in Equation (10). Namely $N_{b}$ bats can capture prey and suck blood eventually with the remaining $N_{p}-N_{b}$ preys escaped and survived:(10)$$B_{N_{p},N_{p}}^{t}=[\begin{array}{c} \begin{array}{ccc} B_{1,1}^{t} & \cdots & B_{1,N_{p}}^{t}\\ \vdots & \ddots & \vdots \\ B_{N_{b},1}^{t} & \cdots & B_{N_{b},N_{p}}^{t} \end{array}\\ \_\\ \begin{array}{ccc} 0\; & \cdots \; & 0\\ \vdots \; & \ddots \; & \vdots \\ 0\; & \cdots \; & 0 \end{array} \end{array}].$$

The reason for augmenting $B_{N_{b},N_{p}}^{t}$ with zero elements is that we try to accomplish symmetric assignment by adding virtual prey or bats in the competition process, but these prey and bats cannot really exist in reality, namely during the process of competition, the $N_{b}-N_{p}$ virtual preys added will become the best prey for $N_{b}-N_{p}$ bats once $N_{b}$ is greater than $N_{p}$, and $N_{p}-N_{b}$ virtual bats will also participate in competing for $N_{p}-N_{b}$ preys once $N_{p}$ is greater than $N_{b}$. However, the real bats cannot take any advantage from the virtual prey and the real prey will not be captured by the virtual bat after the end of assignment. If non-zero values is added instead of zero elements for the weighted maximum matching problem of asymmetric bipartite graph where the elements of $B_{N_{b},N_{p}}^{t}$ are all positive, some virtual bats or preys will be actually assigned to the actual prey or bat once the value of an element added is greater than the minimum element of $B_{N_{b},N_{p}}^{t}$, which will affects the assignment result obviously. Similarly, a very large number (greater than the largest element of the efficiency matrix) should be used to replace the zero element to achieve the same purpose once it is a weighted minimum matching problem of asymmetric bipartite graph.

Each bat then explores one of its most favorite prey and is ready to compete for it, which is equivalent to seeking the largest element of each row in $B_{N,N}^{t}$. For instance, the favorite prey for bat $b_{i}$ can be determined by $p_{bestforb_{i}}^{t}\;=\;arg\underset{o\in \left\{1,2,\ldots ,N\right\}}{\max}B_{i,o}^{t},\;N\;=\;\max\left\{N_{b},N_{p}\right\}$.

#### 4.2.2. Predation Competition

Given that the favorite prey of vampire bat is likely to conflict, namely multiple bats compete for the same prey often occurs, which is called the phenomenon of predation conflict, hence they have to start a predator competition. The biological behavior of bats participating in predatory competition for their favorite prey is one of the biological characteristics of vampire bats, which is famous as the egoism.

The prey robbed by bats are denoted as $\Phi_{prey}^{t}$. Taking $p_{\alpha }$ in $\Phi_{prey}^{t}$ as an example, all bats participated in robbing $p_{\alpha }$ are represented as $\Phi_{bat}^{t}$. The updating formula of $b_{i}$’s interest value for $p_{\alpha }$ is:(11)$$I_{i,\alpha }^{t+1}\;=\;I_{i,\alpha }^{t}-\left(\varphi_{1}^{t}-\varphi_{2}^{t}+\varepsilon \right).$$
where $\varphi_{1}^{t}$ and $\varphi_{2}^{t}$ are the maximum and secondary benefits of the bats in $\Phi_{bat}^{t}$ hunting $p_{\alpha }$, respectively. In order to prevent the update failure of $I_{i,\alpha }^{t}$ due to the equality of $\varphi_{1}^{t}$ and $\varphi_{2}^{t}$, we added ε to ensure the update process runs smoothly.

Predation conflicts no longer occur once each bat has the favorite prey, and the vampire bats begin to suck blood from its favorite prey. The amount of blood sucked by $b_{i}$ are denoted as $z_{i}$. the benefits of the whole generation of vampire bat have been maximized, and we call the process from 4.1.1 to 4.1.2 the first part of IVBO. 

#### 4.2.3. Back Feeding

As discussed in Section 4.2.1, few bats will fail to hunt once there are more bats than prey. Universally, not every bat can draw enough blood to sustain life since predatory competition occurs frequently. Vampire bats will starve to death if they can’t suck enough blood for three consecutive nights [39]. Nevertheless, the life of most vampire bats does not end there, as one of the biological characteristics of vampire bats [40], the biological behavior of sharing extra food with hungry vampire bats according to their kinship, which is famous as the altruism.

After the end of predation, vampire bats began to look for a hungry bat for back feeding according to kinship. For example, $b_{j}$ will get the excess blood from $b_{i}$ once the latter is full and the former is hungry and the genes of them are similar enough, which is called a back-feeding condition as shown in Equation (12). Differences in kinship and starvation between $b_{i}$ and $b_{j}$ are measured by $\tau_{1}$ and $\tau_{2}$, respectively:(12)$$\{\;\begin{array}{c} \left|g_{i}-g_{j}\right|<\tau_{1}\\ \;\;\;\;z_{i}-z_{j}>\tau_{2} \end{array}\;,\;\;j\in \left\{1,2,\ldots ,N\right\}$$

Given that there may be more than one vampire bat meeting the back-feeding condition, hence we regard $b_{j}$ as the optimal transfusion target for $b_{i}$ if the attribute value of $b_{j}$ is the largest, which can be calculated by
(13)$$A_{i,j}\;=\;\frac{e^{w_{1}\times \left(z_{i}-z_{j}\right)}}{w_{2}\times \left|g_{i}-g_{j}\right|},$$
where the weights for differences in kinship and starvation between $b_{i}$ and $b_{j}$ are denoted as $w_{1}$ and $w_{2}$.

The back-feeding process will end once no bats need back feeding, and the blood absorption of the whole bat population is effectively balanced and the benefits of each bat have been balanced. We call the process in 4.1.3 the second part of IVBO.

### 4.3. Cellular Grids Dynamic Partitioning Algorithm

The residual energy of nodes and the total energy cost along with the cover effect can be optimized only for a specific number of sensor nodes by SSBCG and IVBO proposed in 4.1 and 4.2, which has mediocre performance when the number of sensors changes. In order to enhance the robustness of the proposed strategy, a cellular grids dynamic partitioning algorithm (CGDPA) is proposed to dynamically adjust the size of the cellular grid based on the actual number of sensor nodes to enhance the coverage effect when the number of sensors changes.

Regarding the sensor’s perceived radius $R_{S}$ as the radius $R_{C}$ of the cellular grid, the minimum number $N_{ub}$ of cellular grids required for stacking Ω seamlessly can be calculated by Equations (5)–(7), which is also the minimum number of sensors required to cover Ω completely.

As shown in Figure 4a, by regarding the sensing area of the sensors as the circumscribed circle of the cellular grids and denoting the radius of the cellular grids as $R_{C}\;=\;R_{lb}$, the minimum number of cellular grids required for stacking Ω seamlessly, which is denoted as $N_{ub}$, can be calculated by Equation (7) once the length and width of Ω and the sensing radius of the sensor are given.

If the number of sensor nodes is equal to $N_{ub}$, the optimal cover effect as shown in Figure 5a can be achieved; if the number of sensor nodes is more than $N_{ub}$, the coverage rate of Ω can be increased to 100% in spite of the remaining $N_{S}-N_{ub}$ sensors being redundant, which is shown in Figure 5b; however, if the number of sensor nodes is less than $N_{ub}$, on the one hand, $N_{ub}-N_{S}$ cellular grids will turn into monitor blind area, on the other hand, the overlap of the sensing area of $N_{S}$ sensors will be redundant, which finally leads to the inferior monitoring effect shown in Figure 6a.

In order to adapt to the circumstances that the number of sensor nodes is not equal to $N_{ub}$, a cellular grids dynamic partitioning algorithm (CGDPA) is proposed to dynamically adjust the radius of the cellular grids according to the actual number of sensor nodes. Specifically, the radius of the cellular grid is dynamically adjusted by:(14)$$R_{C}\;=\;\{\begin{array}{c} \;R_{lb}\;,\;\;N_{S}\geq N_{ub}\\ R_{ub}-\frac{R_{ub}-R_{lb}}{N_{ub}-N_{lb}}\times \left(N_{S}-N_{lb}\right)\;,\;N_{lb}<N_{S}<N_{ub}\\ \;R_{ub}\;,\;N_{S}\leq N_{lb} \end{array}.$$

$R_{C}$ is the radius of the adjusted cellular grid (dependent variable); $R_{ub}$ and $R_{lb}$ are the radius of the cellular grid as shown in Figure 4a,b, where the sensing area of the sensor node is the inscribed and circumscribed circle of the cellular grid, respectively; $N_{lb}$ and $R_{ub}$ are the number of cellular grids calculated by Equations (5)–(7) when $R_{ub}$ and $R_{lb}$ are regarded as the radius of the cellular grid, respectively; $N_{S}$ is the actual number of sensor nodes (the only independent variable).

The size of the cellular grid will be adjusted in the range shown in Figure 4 according to the number of sensor nodes. When the condition of $N_{lb}<N_{S}<N_{ub}$ is satisfied, as shown in Figure 6b, $R_{C}$ will be increased with the decrease of $N_{S}$, the distance between sensor nodes will be increased, and the redundant coverage area will be effectively reduced; when the critical condition $N_{S}\;=\;N_{ub}$ is met, the optimal cover effect can be achieved without adjusting the size of the cellular grid; when the $N_{S}>N_{ub}$ is satisfied, the coverage rate can be increased to 100% in spite of few sensors being redundant, hence there is no need to adjust the cellular grid size.

### 4.4. Energy-Efficient Coverage Enhancement Strategy for WSNs

The SSBCG, IVBO and CGDPA proposed in Section 4.1, Section 4.2 and Section 4.3 are combined into an energy efficient coverage enhancement strategy for WSNs, which is called improved vampire bat optimizer based on dynamic cellular grids (IVBODCG), and its flowchart is shown in Figure 7. After the amount, the sensing radius, the position of sensors, and the size of the monitoring area are initialized, the first step is to calculate the radius of the cellular grid which is suitable for the actual number of sensors, and divide the monitoring area into cellular grids according to SSBCG, then calculate the distance matrix and expand it based on the relationship between the actual number of nodes and cellular grids, which corresponds to Section 4.4.1. The second step which corresponds to Section 4.4.2 is to find the best mobile destination for sensors by regarding the centroid of cellular grids as the mobile destination of them. The sensor nodes compete for the mobile destination until the best mobile destination of sensors no longer conflicts in the third step, which corresponds to Section 4.4.3. The fourth step calculates the mobile task of the sensors after the end of the competition, then judging whether there are any exchangeable mobile tasks according to the Theorem of task exchange and selecting the most suitable sensor, and it corresponds to Section 4.4.3. The final redeployment task will be carried out once there is no longer any exchangeable task. The detailed process of each section is as follows.

#### 4.4.1. Calculating the Radius of Cellular Grids and the Distance Matrix

$N_{C}$ and $R_{C}$ can be calculated by Equations (5)–(7) and (14), respectively, based on the relationship between the actual number of nodes and $N_{ub}$. The distance matrix $D_{N_{S}\times N_{ub}}$ can be calculated by Equation (2). Then we regard the sensors and cellular grids as vampire bats and as prey, respectively.

The number of bats is equal to that of prey when $N_{S}=N_{ub}$ is satisfied, which means $D_{N_{S}\times N_{ub}}$ is a square matrix, which can be transformed into a symmetric assignment problem. The number of bats is not equal to that of prey when $N_{S}=N_{ub}$ is not satisfied, which means $D_{N_{S}\;\times \;N_{ub}}$ is no longer a square matrix and it belongs to the asymmetric assignment problem, and it can be transformed into a symmetric assignment problem by adding $N_{S}-N_{ub}$ virtual cellular grids when $N_{S}>N_{ub}$ and adding $N_{ub}-N_{S}$ virtual sensors when $N_{S}<N_{ub}$, which is equivalent to adding zero elements of $N_{S}-N_{ub}$ columns to $D_{N_{S}\times N_{ub}}$ and extending it to a square matrix when $N_{S}>N_{ub}$, and adding zero elements of $N_{ub}-N_{S}$ rows to $D_{N_{S}\times N_{ub}}$ and extending it to a square matrix when $N_{S}<N_{ub}$.

Given that IVBO is used to optimize the total energy cost of sensors, which can effectively solve the maximum matching problem when considering only the predation competition of vampire bats. However, the purpose of reducing the total energy cost of nodes is to find the minimum value. Therefore, the benefit matrix should be calculated by $B_{N\times N}\;=\;-D_{N\times N}$. We then proceed to Section 4.4.2.

#### 4.4.2. Seeking the Optimal Moving Destination for Sensors

We traverse all sensors and cellular grids to seek the optimal moving destination for each sensor. We define $G_{j}$ as the best moving destination for $S_{i}$ if $G_{j}$ is closest to $S_{i}$. The optimal moving destination for $S_{i}$ on the t-th iteration is calculated by $G_{bestfors_{i}}^{t}=arg\underset{j\in \{1,2,\ldots ,N\}}{\max}B_{i,j}^{t}$, where $B_{i,j}^{t}$ is the benefits of $S_{i}$ when moves to $G_{j}$ on the t-th iteration.

We determine whether the optimal moving destination of the sensors is in conflict; if there is no conflict, then the moving task matrix can be calculated by $Task_{N\times N}^{t}\;=\;-D_{N\times N}^{t}⨂X_{N\times N}^{t}$, where $D_{N\times N}^{t}$ is the distance matrix calculated in 4.4.1, and the meaning of $x_{i,j}^{t}\in X_{N\times N}^{t}$ is shown in Equation (4). We then proceed to Section 4.4.4; if there is a conflict, then proceed to Section 4.4.3.

#### 4.4.3. Competition

When multiple sensor nodes compete for the same cellular grid, we denote the conflicting cellular grid robbed by multiple nodes as a popular grid set $\Phi_{girds}^{t}$. For instance, taking $G_{\alpha }^{t}$ in $\Phi_{girds}^{t}$ as an example, sensors participated in robbing $G_{\alpha }^{t}$ are represented as $\Phi_{sensors}^{t}$. The updating formula of $S_{i}$’s benefit in moving to $G_{\alpha }^{t}$ is:(15)$$B_{i,\alpha }^{t+1}\;=\;B_{i,\alpha }^{t}-\left(\varphi_{1}^{t}-\varphi_{2}^{t}+\varepsilon \right),$$
where $\varphi_{1}^{t}$ and $\varphi_{2}^{t}$ are the maximum and secondary benefits of the nodes in $\Phi_{sensors}^{t}$ moving to the $G_{\alpha }^{t}$, respectively. In order to prevent the update failure of $B_{i,\alpha }^{t}$ due to the equality of $\varphi_{1}^{t}$ and $\varphi_{2}^{t}$, we added ε to ensure the update process runs smoothly. All popular grids and the conflicted sensors will be traversed to update the benefit matrix $B_{N\;\times \;N}^{t}$. Then return to Section 4.4.2.

#### 4.4.4. Exchanging the Moving Tasks

So far, the benefits of all sensors have been maximized but the differences have not been minimized. We traverse all nodes to exchange their moving tasks based on the theorem of task exchange; then proceed to Section 4.4.5: Condition of task exchange: Given that sensors $S_{i}$ and $S_{m}$ with $G_{j}$ and $G_{n}$ as their moving destination, the task exchange condition is:(16)$$\{\begin{array}{c} d_{i,n}^{t}<d_{i,j}^{t}\\ d_{m,j}^{t}<d_{i,j}^{t} \end{array}.$$Lemma of task exchange: The benefit of $S_{i}$ and $S_{m}$ can be balanced by Equation (17) once they satisfy the condition of task exchange:(17)$$\{\begin{array}{c} x_{i,n}^{t}\;=\;1\\ x_{m,j}^{t}\;=\;1\\ x_{i,j}^{t}\;=\;0\\ x_{m,n}^{t}\;=\;0 \end{array}$$Theorem of task exchange: n schemes can be found once n sensors satisfying the condition of task exchange simultaneously based on the lemma of task exchange. We define $S_{\xi }$ as the optimal exchangeable sensor for $S_{i}$ once it has the largest fitness value among n sensors, which can be calculated by:(18)$$fit_{\xi }\;=\;\frac{e^{-\mu_{1}\times \left(Task_{i,n}^{t}+Task_{\xi ,j}^{t}\right)}}{\mu_{2}\times \left|Task_{i,n}^{t}-Task_{\xi ,j}^{t}\right|}.$$
where $\mu_{1}$ and $\mu_{2}$ are the weights of the summary and difference of the benefit of them, respectively.

For $S_{i}$ and its moving destination $G_{j}$ shown in Figure 8, both $S_{m_{1}}$ and $S_{m_{2}}$ satisfy the condition of task exchange. The fitness value of $S_{m_{1}}$ and $S_{m_{2}}$ can be calculated by Equation (18), and the former is bigger than the latter; thus, we exchange the moving task of $S_{i}$ and $S_{m_{1}}$ rather than that of $S_{i}$ and $S_{m_{2}}$.

Taking a Ω of 50 m×50 m as an example, when calculating the movement scheme of 42 sensors with a perceived radius of 5.25 m, the comparison before and after the task exchange is shown in Figure 9, and it can be seen that the distant moving tasks as shown by red arrows in Figure 9a are balanced in Figure 9b, which is shown by green arrows.

#### 4.4.5. Redeployment

The energy-efficient redeployment can be completed by moving sensors according to the matrix $Task_{N\;\times \;N}$.

## 5. Simulation Results

### 5.1. Parameter Setting

Simulations were performed with MATLAB R2019a on a computer with a 2.7 GHz frequency and 8 GB memory to evaluate the performance of our proposed strategy IVBODCG, and the MATLAB code is detailed in the Appendix A. We compared the performance of VFA, VFPSO, LGWO and IVBODCG about final coverage rate (FCR), total energy cost (TEC) of all sensors, uniformity of residual energy (URE) and maximum energy cost (MEC) of sensors under the same experimental conditions, which are shown in Table 1.

### 5.2. Simulation Results

The differences in final locations, moving trajectories, and cover effect of sensor nodes by four algorithms are presented as Figure 10. 

The figures from (a1) to (d1) show the initial locations of nodes for algorithms for fairness of comparison, and it can intuitively show that our proposed strategy IVBODCG is superior than VFA, VFPSO and LGWO when considering the actual moving distance, the final coverage effect, and the uniformity of the mobile distance of sensors.

IVBODCG reaches an FCR of 100% while that of LGWO, VFA and VFPSO are only 92.70%, 93.56% and 95.44%, which is compared in Figure 11a. Since there are obvious differences in FCR by the end of redeployment, the TEC is compared in Figure 11b when four algorithms all reach a coverage rate of 92.70%. The TEC of VFPSO, LGWO and VFA is 1.8306 × 10^4^, 2.8991 × 10^4^ and 2.2132 × 10^4^ Joules when the coverage rate of them all reach 92.07%, respectively, and they are worse than that of IVBODCG by 49.39%, 136.58% and 80.61%, and the former even consume more energy when the latter achieves full cover effect. In addition, IVBODCG achieved an FCR of 100% with a TEC of 1.6148 × 10^4^ Joules while the coverage rate of LGWO, VFA and VFPSO only reached 80.6%, 85.8% and 89.2%, respectively, with the same energy cost.

The energy cost of each sensor of IVBODCG, VFPSO, VFA and LGWO are compared in Figure 12, LGWO has the worst performance when considering TEC of nodes, and VFPSO along with VFA are slightly better than LGWO, and IVBODCG are obviously better than all of them. In addition, URE and MEC of nodes of IVBODCG is effectively optimized when compared with other three algorithms, as a result of that they only care about the cover effect and the convergence speed with the optimization of energy cost ignored.

The performance differences of MEC of nodes of four algorithms are presented in Figure 13a. The node with the maximum energy cost of IVBODCG, VFPSO, VFA and LGWO reaches the optimal location after 13, 17, 29 and 33 rounds of deployment, and consume 590.83, 829.08, 1406.32 and 1533.47 Joules of energy, respectively. In addition, URE of IVBODCG, VFPSO, VFA and LGWO are 144.51, 183.29, 261.56 and 350.69 Joules after the final rounds of movements, which are shown in Figure 13b. It indicates that VFPSO, LGWO and VFA performs worse than IVBODCG by 26.84%, 142.6% and 81.00%, respectively. Unfortunately, our strategy is worse than the other three algorithms during the first 10 rounds of movement. Given that URE is related to the difference in distance moved by each sensor, namely, the higher the uniformity of residual energy, the greater the difference of moving distance of each node during the movement since the actual moving speed by single step is the same.

Obviously, the sensor nodes driven by each algorithm have not all moved to the destination by the end of the 10th round of movement, hence the reason why the URE of IVBODCG during the first 10 rounds is higher is that a large number of sensor nodes have reached the best destination early while other nodes are still moving, and the sensor nodes of the other three algorithms have not arrived the destination and are moving at a speed of 1 m by single step, and it is consistent with the phenomenon shown in Figure 10d2, which shows that our proposed strategy IVBODCG has a large number of sensor nodes with short moving distances when compared to VFPSO, VFA and LGWO.

By initializing different positions for sensors, 200 independent simulation experiments are performed to evaluate the stability and reliability of IVBODCG. The initial positions of the sensors were randomly generated, and the other parameters were the same in every simulation experiment. The performance comparison about FCR of IVBODCG, VFA, VFPSO and LGWO is shown in Figure 14a, which can be seen that LGWO is the worst one that floating around 92%, and VFA is close to 93% which is slightly superior than LGWO. Our proposed strategy can reach an FCR of 100% by every single experiment while VFPSO can only reaches 95%. Figure 14b compares the TEC of the four algorithms after completing the moving task, which is consistent with the experiment result presented in Figure 12. VFPSO, LGWO, and VFA fluctuate around 2.0 × 10^4^, 2.8 × 10^4^ and 2.5 × 10^4^ Joules, respectively, while IVBODCG is close to 1.6 × 10^4^ Joules. Figure 14c,d compare the performance differences in MEC and URE, and our proposed strategy is the best among them.

The mean value of 200 simulation experiments are presented in Table 2. IVBODCG can balance the URE of sensors by 48.36%, 41.51% and 24.73%, and also reduce the MEC of nodes by 48.66%, 41.98% and 24.94%, when compared to LGWO, VFA and VFPSO. Besides, it can reduce the TEC of nodes by 42.03%, 34.73% and 18.25%, and also have a superior performance when considering the FCR.

Given that conclusions based on the means are generally misleading, and it is hard to find whether there is any statistically significant difference between the approaches without the indication of error, the difference in the statistical results of the performance of the four algorithms in each performance indicator is presented in Figure 15 and Table 3. 

Considering that the actual number of sensors may not be the optimal number which is 53, a vertical comparison is made by changing the number of sensors in order to test the universality of our proposed strategy. Figure 16 compares the performance of four algorithms when the number of sensors varies from 30 to 80. It can be seen from Figure 16a that the FCR of the four algorithms is increasing with the increase of the number of sensors, and our proposed strategy is superior than that of LGWO and VFA when the number of sensors is the same, and FCR of IVBODCG reaches 100% when the number of sensors exceeds 53. Figure 16b shows that TEC of LGWO, VFA and VFPSO increases obviously with the increase of the number of sensors, and that of our strategy is slightly increasing before the number of nodes increases to 53, and is lower than that of LGWO, VFA and VFPSO when the number of sensors is the same. Figure 16c,d indicate that the MEC and URE of sensors of LGWO and VFA is gradually increasing with the increase of the number of sensors, and that of VFPSO is slightly rising while IVBODCG is constant and even gradually reduced when the number of sensors is greater than 53, and it also has superior performance when the number of sensors is the same.

Surprisingly, the TEC, MEC and URE of IVBODCG are almost constant or slightly increasing when the number of sensors is less than 53, and are decreasing rather than increasing as the number of sensors increases beyond 53 as shown in Figure 16b–d, which is different from the other three algorithms. However, this is not without reason, which is precisely thanks to the virtual grids and virtual sensors introduced in Section 4.4.1. The TEC, MEC and URE of sensors are almost constant or slightly increasing when the number of sensors is less than 53, which is due to the game theory of IVBO, namely it can always find a scheme to optimize the moving distance. Regardless of the number of sensors, it can assign sensor nodes to the cellular grid in close proximity. Analogously, when the number of sensors is greater than 53 and continues to increase, the number of cellular grids produced by CGDPA and SSBCG will stay at 53 instead of increasing, namely the number of sensors will be greater than the number of cellular grids, which causes the assignment problem to become asymmetrical. However, the existence of virtual grids transforms it into a symmetric assignment problem, and some of the sensor nodes will correspond to these virtual grid points, and they are assigned only in the IVBO instead of moving actually during redeployment. Therefore, IVBO obtains the rights of option for picking 53 nodes out of all sensor nodes to move, rather than moving all of them. As the number of sensors continues to increase, the sensors that are extremely close to cellular grids, predictably, can be found for redeployment, hence the moving distance so as to the TEC, MEC and URE is optimized.

## 6. Discussion

As the derivative algorithms of VFA, such as the strategies proposed in [8,9,10,11], their basic principle is to fill unmonitored areas and separate overlapping nodes. During each iteration, the virtual moving effect of a sensor is influenced by the threshold of the repulsion between the nodes, the threshold of the attraction from grid points to sensors, and the single-step moving speed. Therefore, the threshold of repulsion, attraction and the virtual moving speed by single-step have a significant impact on FCR; the optimal coverage effect will not be obtained if the best parameters cannot be determined. In general, VFA-based approaches involve a number of parameters to decide the magnitude of the force and to prevent sensors from oscillation, which means, that it is usually difficult to find the suitable values for these parameters in different cases. In addition, just similar to other swarm intelligence, PSO and GWO save the calculation time by sacrificing the accuracy of the solution, hence the convergence speed and the performance of the convergence value are poor for large scale problems, for example, optimizing the coverage effect. Conversely, the performance of IVBO is insensitive to its parameters thanks to it transforming the sensor coverage enhancement problem to a task-assignment problem, it ensures an optimum coverage rate after the movements, which is the reason that FCR of VFA, LGWO and VFPSO are lower than that of IVBODCG in 200 simulation experiments as shown in Figure 14 and Figure 15 and Table 2 and Table 3. Although the parameters of the former are adjusted repeatedly, the effect cannot be improved significantly.

The virtual movements of nodes in each iteration, similarly, the moving path of the nodes and the moving energy consumption, are affected by the defects of VFA and its derivative algorithms, which means it is difficult to account for the optimization of TEC, MEC and URE of sensors by VFA and VFPSO, not to mention LGWO. The task distributing model of IVBO results in that the first part of it is to optimize TEC on the basis of a full coverage rate. Therefore, IVBODCG has a superior performance in terms of energy cost, which is shown in Figure 11 and Table 2.

The first part of IVBO can minimize the TEC of sensors while ensuring the best coverage, just as it can optimize the benefits of the generation of vampire bats. However, the significances of the IVBO are far greater. The reverse blood-transfusion process in the second part of IVBO guarantees the balance volume of blood sucked by vampire bats, which means the moving task of a nodes with a long distance of movement is exchanged by another sensor. Thus, IVBODCG is undoubtedly superior than VFA, VFPSO, and LGWO when considering MEC and URE of sensors. As an example, energy-efficient coverage enhancement problem is merely one of the numerous applications of IVBO, and it may perform better than the general integer programming method when solving a large category of task assignment problems with the goal of equilibrium.

## 7. Conclusions

In this paper, by using cellular grids to stack the sensing area seamlessly, and the optimization problem of coverage enhancement and energy consumption is converted into a task distributing problem. In addition, CGDPA is proposed to improve the coverage effect for different numbers of sensors. Furthermore, IVBO is presented to tackle the asymmetric competition problem by introducing virtual bats and virtual preys, which solves not only the multi-objective problem of minimizing and balancing the energy consumption of sensor nodes, but also the asymmetric assignment problem when the number of sensor nodes is not equal to that of cellular grids. We combine SSBCG, CGDPA and IVBO that proposed into an energy-efficient coverage enhancement strategy IVBODCG for WSNs. Simulation results show that, compared with three classical algorithms, the strategy proposed shows improved performance in terms of FCR, TEC, MEC and URE, and it also has a superior robustness when the number of nodes changes. However, there are some limitations, for example, the same perception radius of sensor nodes and the disk coverage model, which are too simplistic and ideal to be used in realistic applications, some other related coverage models such as confident information and data fusion based coverage model which define coverage concept from the view of reconstruction and estimation should be considered. Additionally, some true experiments instead of theoretical simulations of IVBO in WSNs will be our research focus in future.

## Figures and Tables

**Figure 1 sensors-20-00619-f001:**
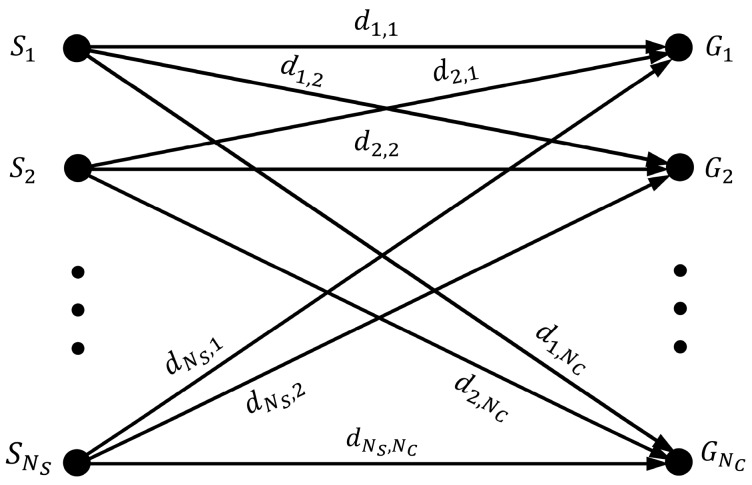
Bipartite graph model for the task distributing problem of assigning $N_{C}$ mobile destinations for $N_{S}$ sensor nodes. $\left\{S_{1,}\;S_{2},\;\ldots ,\;S_{N_{S}}\right\}$ and $\left\{G_{1,}\;G_{2},\;\ldots ,\;G_{N_{C}}\right\}$ represent the set of $N_{S}$ sensors and $N_{C}$ grid points, respectively. The weight of the edge $<S_{i},G_{j}>$ denotes the distance between $S_{i}$ and $G_{j}$
_._

**Figure 2 sensors-20-00619-f002:**
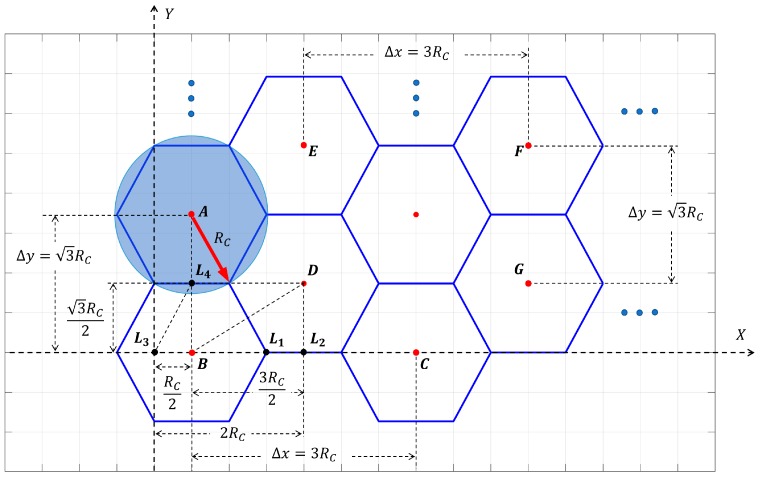
Stacking diagram: the reference cellular grids and the stacking interval when stacking a given area seamlessly with cellular grids.

**Figure 3 sensors-20-00619-f003:**
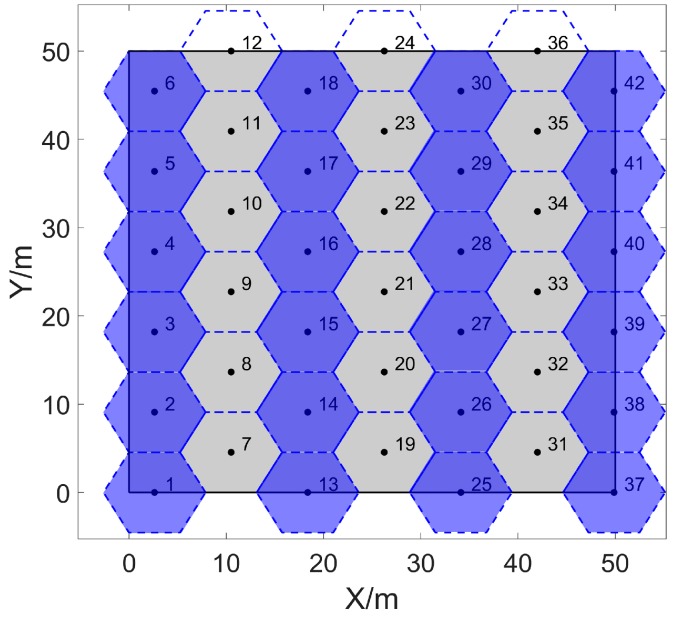
Stacking the two-dimensional area seamlessly with 42 cellular grids. The length and width of the area are both 50 m, and the radius of each cellular grid is 5.25 m.

**Figure 4 sensors-20-00619-f004:**
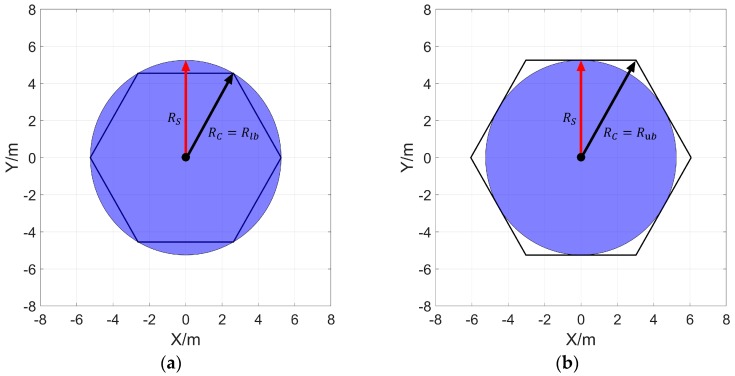
The relationship between the cellular grids and sensor sensing range. (**a**,**b**) show that the sensor’s sensing range is the circumscribed and inscribed circle of the cellular grid.

**Figure 5 sensors-20-00619-f005:**
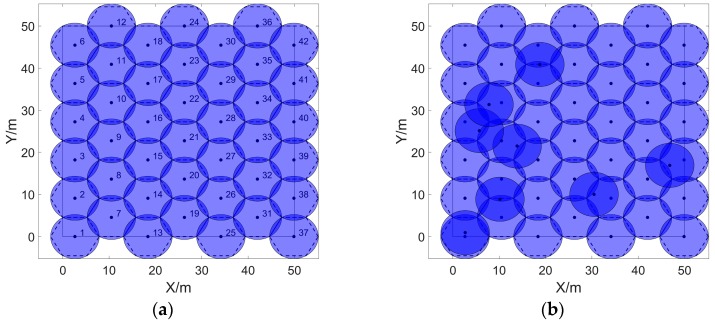
Coverage effects of sensors. (**a**) shows that the sensing area is completely covered when it satisfies the condition $N_{S}\;=\;N_{ub}$; (**b**) shows that the sensing area is completely covered and 8 sensors are redundant when using 50 sensors to cover 42 cellular grids.

**Figure 6 sensors-20-00619-f006:**
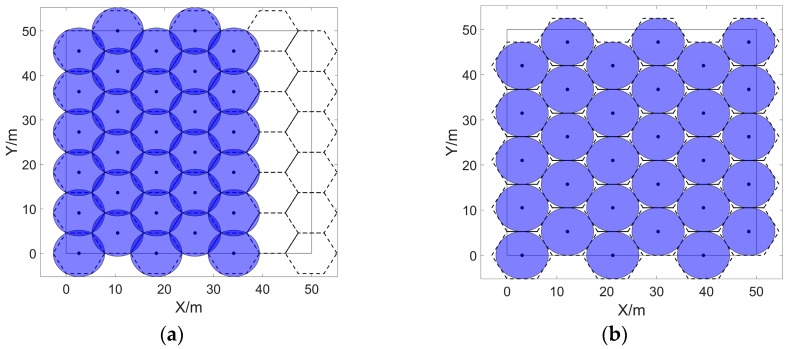
Coverage effects of sensors. (**a**) The sensing area is incompletely covered, and 12 cellular grids become the monitor blind area when the condition $N_{S}<N_{ub}$ is satisfied and the radius of cellular grids is not adjusted. (**b**) The problem of the monitoring blind area is effectively improved by adjusting the radius of cellular grids.

**Figure 7 sensors-20-00619-f007:**
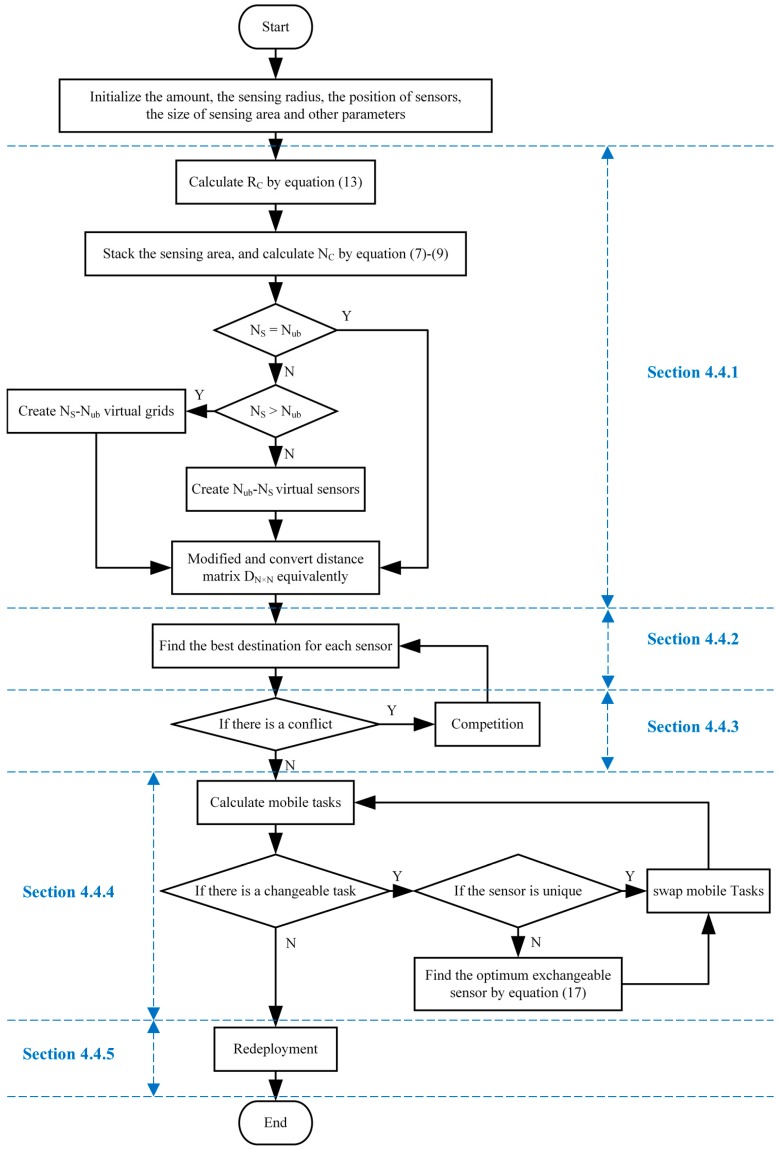
The flow chart of the energy-efficient coverage enhancement strategy for WSNs.

**Figure 8 sensors-20-00619-f008:**
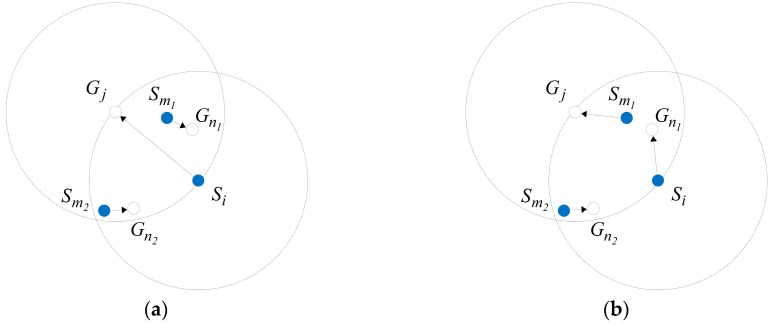
Schematic diagram of task exchange when multiple sensors meet the task exchange conditions. (**a**,**b**) are the schematic diagrams before and after task exchange, and the optimal exchange scheme of $S_{i}$ is determined by the theorem of task exchange.

**Figure 9 sensors-20-00619-f009:**
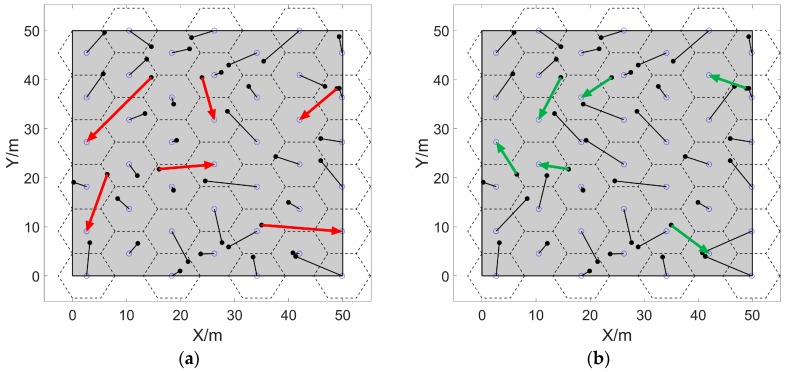
Comparison of moving trajectories before and after task exchange. (**a**,**b**) are the moving trajectories before and after the task exchange. The hollow and the solid points are the final and initial positions of nodes, respectively. The lines connecting them are the moving trajectories of sensors.

**Figure 10 sensors-20-00619-f010:**
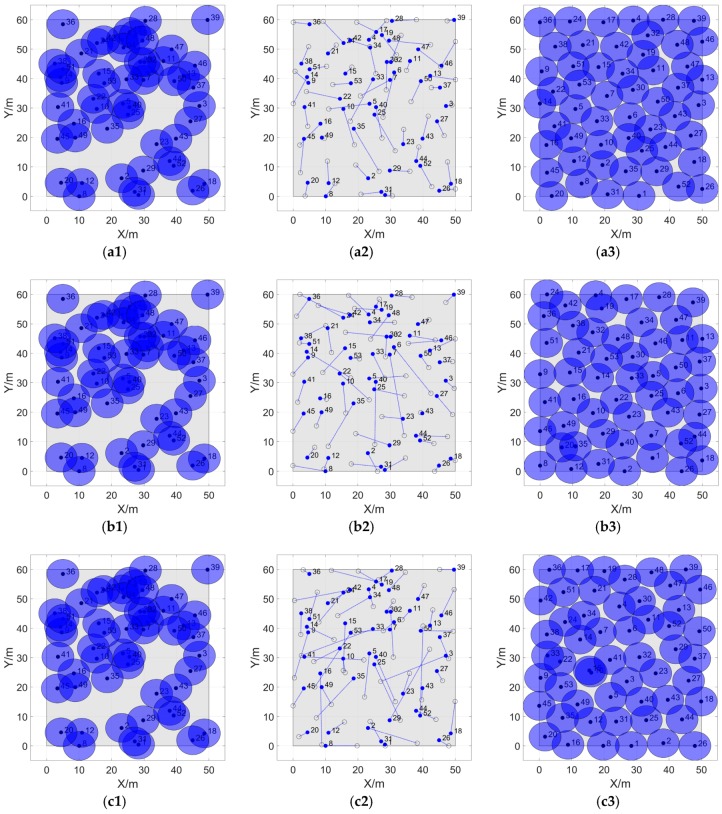
Coverage enhancement effects and movement trajectories of VFPSO, VFA, LGWO and IVBODCG after 28 rounds of movement. The circular areas with the solid points at the center are the perceived ranges of sensors. (**a1**) to (**d1**) are the initial positions of the four algorithms, and (**a3**) to (**d3**) are the final coverage effects of them. The 53 hollow and solid points in (**a2**) to (**d2**) are the final and initial locations of 53 nodes, respectively. The lines connecting them are the actual mobile trajectories of sensors.

**Figure 11 sensors-20-00619-f011:**
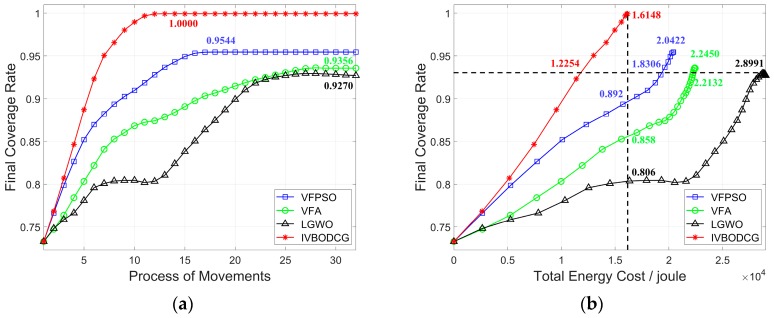
Coverage rates and energy cost of four algorithms. (**a**) shows the relationship between final coverage rate and process of movements. (**b**) shows the relationship between final coverage rate and total energy cost. Compared with VFA, VFPSO and LGWO, IVBODCG can achieve the optimal coverage effect with the least energy cost and the fewest rounds of movements.

**Figure 12 sensors-20-00619-f012:**
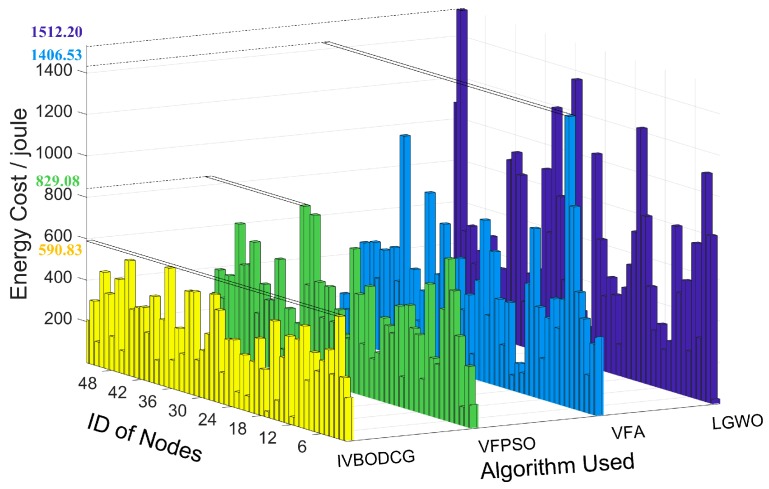
Energy cost for 53 sensors of IVBODCG, VFPSO, VFA and LGWO. The yellow, green, light blue, and dark blue bar graphs are the energy cost of the 53 nodes of four algorithms, respectively, after redeployment. The bars from right to left in a given bar graph represent the energy cost of nodes 1–53 of the corresponding algorithm.

**Figure 13 sensors-20-00619-f013:**
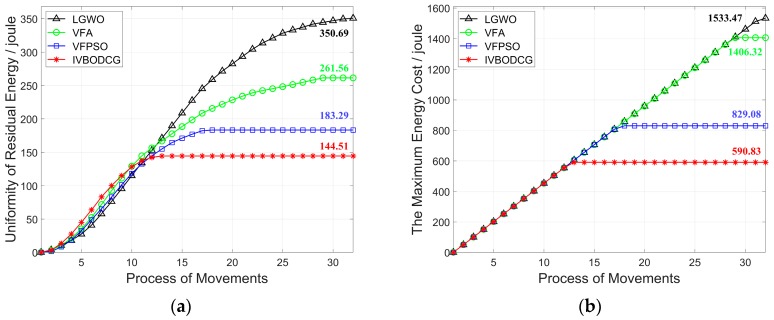
Comparison of MEC and URE of nodes of four algorithms. (**a**) shows the relationship between uniformity of residual energy and process of movements. (**b**) shows the relationship between the maximum energy cost and process of movements. Compared with LGWO, VFA and VFPSO, the IVBODCG’s MEC and URE is the best after the final round of movement.

**Figure 14 sensors-20-00619-f014:**
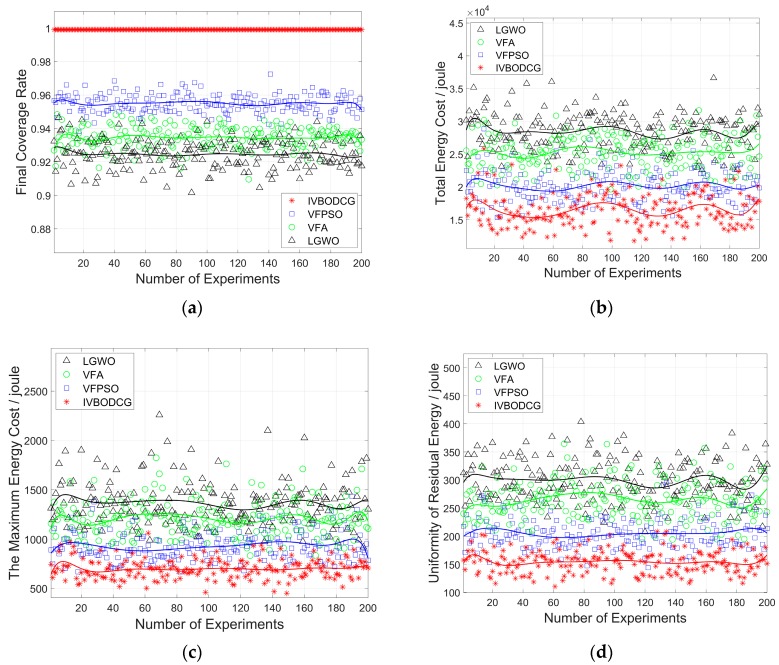
Comparison of energy cost and final coverage rate by 200 independent experiments. (**a**–**d**) show the final coverage rate, total energy cost, the maximum energy cost and uniformity of residual energy of four algorithms in 200 independent experiments, respectively. The curve is the fitting result of the tenth polynomial of the data points.

**Figure 15 sensors-20-00619-f015:**
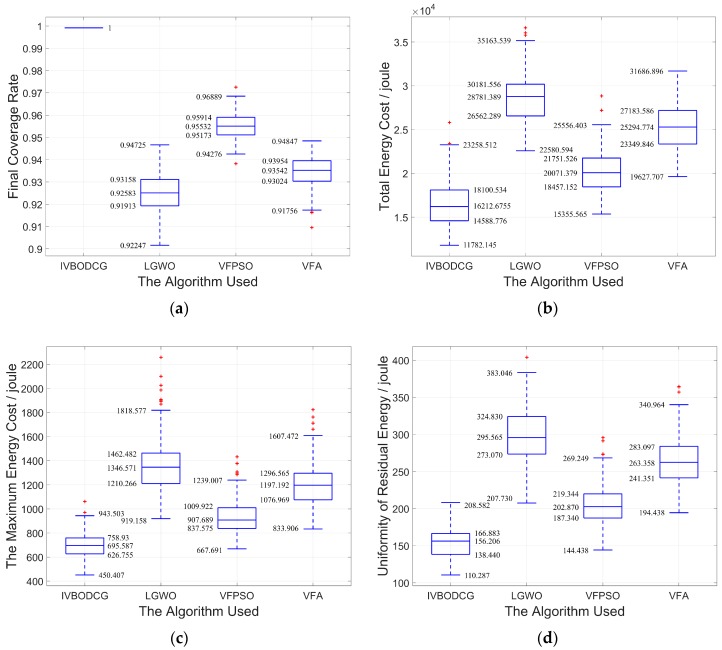
Comparison of statistical results of the four algorithms from 200 experiments. (**a**–**d**) show the statistical results of final coverage rate, total energy cost, the maximum energy cost and uniformity of residual energy of four algorithms, respectively.

**Figure 16 sensors-20-00619-f016:**
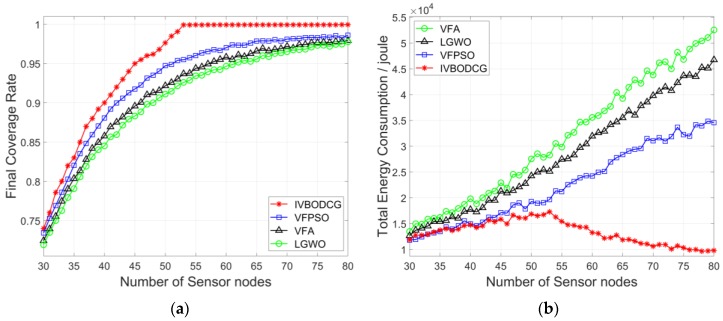
Comparison of four algorithms when the number of sensors changes. (**a**–**d**) show the final coverage rate, total energy cost, the maximum energy cost and uniformity of residual energy of four algorithms, respectively.

**Table 1 sensors-20-00619-t001:** Simulation parameters.

Parameters	Value
Length of sensing area	60 m
Width of sensing area	50 m
Perceived radius of sensors	5 m
Number of grids (to calculate coverage)	601 × 501
Number of sensors	30~80
Single step moving distance during actual movement	1 m
Energy cost by moving unit distance	50.4 J/m
Radius of cellular grids	5 m~5.72 m
Initial energy of sensors	3000 Joules
Distance threshold of virtual force	8.58 m
Weight of TEC	0.2
Weight of URE	0.8
Updating factor	0.1
Maximum number of iterations	100
Amount of wolfs	20
Number of particles	20
Inertial factor	0.8
Individual cognitive coefficient	2
Global cognitive coefficient	2

**Table 2 sensors-20-00619-t002:** Performance comparison of 200 simulation experiments with the average results.

Performance Index	LGWO	VFA	VFPSO	IVBODCG
FCR	92.47%	93.46%	95.51%	100%
TEC/Joules	28,447.5	25,266.8	20,172.1	16,490.5
MEC/Joules	1362.8	1205.8	932.1	699.6
URE/Joules	299.4	264.3	205.4	154.6

**Table 3 sensors-20-00619-t003:** Performance comparison of 200 simulation experiments in standard deviation.

Performance Index	LGWO	VFA	VFPSO	IVBODCG
FCR	0.85%	0.66%	0.54%	0.00%
TEC/Joules	2742.1	2537.6	2333.1	2533.9
MEC/Joules	226.1	181.6	144.4	102.6
URE/Joules	36.8	33.4	26.3	20.5

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
