# Peer review of "An Energy-Efficient Coverage Enhancement Strategy for Wireless Sensor Networks Based on a Dynamic Partition Algorithm for Cellular Grids and an Improved Vampire Bat Optimizer"

_sensors, 2020, doi:10.3390/s20030619_

Round 1

Reviewer 1 Report

the article has been improved

Author Response

We thank you for your appreciation of our contributions.

Reviewer 2 Report

Coverage is one of the crucial research issues in wireless sensor networks, which reflects the network quality of service. The authors in this manuscript focus on the coverage problem in wireless sensor networks. The authors have proposed an energy-efficient coverage enhancement strategy for WSNs by combining SSBCG, CGDPA and IVBO. The effectiveness and efficiency of the proposed algorithm is also verified by simulation experiments. The discussed topic is within the scope of this journal, and the organization of this manuscript is good.

However, there are a series of concerns should be addressed before this manuscript can be further processed by this journal.

Why don’t the authors give the proposed strategy a name? The authors like to use long sentences, but many of them are not clearly expressed logically, which seriously affects the readability. For example, this sentence in the abstract “In this paper, by splicing the sensing area optimally by cellular grids, the coordinates of the best deployment location and the calculation strategy of the required minimum number of nodes are revealed once the perceived radius of nodes are given, and the optimization problem of maximizing coverage and minimizing and equalizing energy consumption is converted into a task distributing problem of assigning mobile destinations for sensor nodes, and a dynamic partition algorithm for a cellular grid is also proposed to improve the coverage effect for different numbers of sensors” The proposed coverage enhancement strategy is only based on the binary sensor coverage model which is usually called Boolean coverage model or disk coverage model in coverage studies in wireless sensor networks. Although the disk coverage model has good geometric symmetry, however, this model is too simplistic and ideal to be used in realistic applications. Some other related coverage models such as confident information coverage model which defines coverage concept from the view of estimation can be considered. There are many grammatical and spelling mistakes in this manuscript, which directly affects the readability. Please proofread and scrutinize the whole manuscript and correct them. The authors review some related works on the methods of coverage enhancement strategy. However, they didn’t survey some up-to-date works including the confident information coverage model-based coverage enhancement strategy, data fusion-based coverage enhancement strategy, etc. Please supplement some up-to-date related works and references on some other novel coverage models.

Author Response

See Annex for details.

Reviewer 3 Report

The paper is well written and similar to other submitted by authors. I have some general concerns; it's an opinion of mine, but I think it is correct. This paper, exceedingly complex, presents a sophisticated algorithm for WSNs. But the coverage of a single sensor is a circle, that is normally not true due to environment interference. There is not a single actual experiment. Sensors have a radius of 100m, too much for true sensors. Everything is theoretical. The software is not given, therefore is not possible even to double check results. Of course decision is up to Associate Editor, but for me two things are mandatory:

some true experiments, showing that all things can be done and are useful to engineers developing WSNs the software must be released as open source

Otherwise this paper is only a theoretical study, basically useless because not repeatable. Sorry.

Author Response

See Annex for details.

Round 2

Reviewer 2 Report

The authors have done a good revision, answered all my questions and addressed all my concerns. Good job.

This manuscript is a resubmission of an earlier submission. The following is a list of the peer review reports and author responses from that submission.

Round 1

Reviewer 1 Report

The article describes a new algorithm in the WSN network enabling better management of node energy and their distribution in order to cover the monitoring area.

In my opinion  the structure of article is correctly but results and conclusions are not very convincing.
Your algorithm (VBO) is presented well and I understand how it works, but you compare it with algorithms that are very unpopular. Especially SRACC - I found only one article where this algorithm was described. Algorithms based on gray wolf optimization are more but the VFLGO version is not very popular.
I don't know why you chose these algorithms - justify it. Explain how they work in brief. In addition, you specify how they were implanted in matlab

It is best if it gives a comparison your algorithm with classic algorithms for wsn.

Units - in text, for dystance you used feet, on figure meters, chang all for meters

References - references are quite good chosen and new, but on 25 position 16 is authored by scientists from China. This suggests that you only know scientific solutions from China
In my opinion, you should add more international authors and articles.

END

Reviewer 2 Report

I consider two important things missing, which would enhance the outcome of the manuscript.

Firstly, the author claim that they have performed 200 experiments with randomly generated positions of the sensors. The simulation results include neither standard deviation nor confidence intervals or some other indication of error. This information is essential to the correct interpretation of results because conclusions based on the means only are generally misleading. Without the error bars or confidence intervals, it is hard to see whether there is any statistically significant difference between the approaches under test.

Secondly, energy modelling is a crucial element in wireless network simulation, and energy consumption is an essential metric for evaluating the performance of wireless network protocols. The authors provide only the simulation parameters in Table 3, but I cannot see which energy model has been used. The authors should briefly present the energy model used in this paper and explain why the particular model has been chosen.

Reviewer 3 Report

In this paper, the authors proposed a dynamic partition algorithm to improve the network coverage, and a vampire bat optimizer is proposed for the optimization of network energy consumption. The topic is itself interesting and adequate for a journal, but a lot of issues still exist and need to fix them, such as:

Please rewrite the abstract lines from 23 to 28 and from 48 to 50 to make it more clear. Please enhance the introduction part. The plagiarism of related works (2) part is very high. Please rewrite the whole 2nd part (related works). Explanation of centroid coordinates (equation 10 & 11) is not enough. Please add more detail. The relationship between a vampire bat and WSN behavior is not clear. Please add more detail to make it more effective before proposing the vampire bat optimizer. Please explain in detail to Dynamic Partitioning Algorithm (Section 4.3). Use any standard software for writing the equation/mathematical work.  What do you mean by “Set the number” in the flowchart (Fig. 8). Explain the flowchart of the energy-efficient coverage enhancement strategy in detail. Please add the detail of the simulator at the starting of section 5 (simulation results section).  Fig. 12 (a) shows the VBO balance degree of residual energy is greater in the starting (from 3 to 10), please explain why it is greater in the case of VBR? In Fig. 13(a), VBO coverage rate is varying too much between 29 to 42 numbers of sensors. Please explain the reason? In Fig. 13(b), the energy consumption of VBO is looking like constant where the numbers of sensors are increasing, which is not possible. Can you please give more detail to make it clear? Please add more references (at least 3 to 4) related to the “energy-efficient coverage enhancement algorithm for wireless sensor networks.” The English language must be improved.

Round 2

Reviewer 1 Report

The authors corrected the article in line with my comments
The introduction has been changed and expanded
The authors now compare their article with 3 other algorithms more popular than the previous ones.
The results of the comparison are presented correctly
The bibliography has been completed.

I think that a detailed description of the implementation of the VBO algorithm in matlab should be added.
and
Figures 15 is valuable but not very transparent especially in the printed version. Maybe it is better to draw points and a trend line - the authors will check it out.
END

Reviewer 2 Report

In the revised version of the paper, the authors have addressed all my concerns.

Reviewer 3 Report

Right now, the paper plagiarism is 41%

The authors already published the paper (DOI:: 10.1109/JIOT.2019.2952718) in a journal ( IEEE Internet of Things Journal ) with the same algorithms. 

I think at least; authors should add a 60% new contribution in a new journal paper. 

Please justify the new contribution of this paper.